# The longitudinal impact of employment, retirement and disability status on depressive symptoms among men living with HIV in the Multicenter AIDS Cohort Study

**Deanna Ware**[1]*, **Sergio Rueda**[2], **Michael Plankey**[1‡], **Pamela Surkan**[3], **Chukwuemeka N. Okafor**[4], **Linda Teplin**[5], **M. Reuel Friedman**[6‡]

**1** Division of Infectious Diseases, Department of Medicine, Georgetown University Medical Center, Washington, District of Columbia, United States of America, **2** Department of Psychiatry, Institute of Medical Science, Institute of Health Policy, Management and Evaluation, University of Toronto, Toronto, Canada, **3** Johns Hopkins Bloomberg School of Public Health, John Hopkins University, Baltimore, Maryland, United States of America, **4** Department of Public Health, Robbins College of Health and Human Sciences, Baylor University, Waco, Texas, United States of America, **5** Feinberg School of Medicine, Northwestern University, Chicago, Illinois, United States of America, **6** Department of Infectious Diseases and Microbiology, Graduate School of Public Health, University of Pittsburgh, Pittsburgh, Pennsylvania, United States of America

‡ These authors are joint senior authors on this work.
* dpf38@georgetown.edu

**Data Availability Statement:** Data cannot be shared publicly because data contains potentially sensitive information covered by a NIH Certificate

## Abstract

Many persons living with HIV (PLWH) either reduced their employment capacity or stopped work completely due to disease progression. With the advent of effective antiretroviral therapy, some PLWH were able to return to the workforce and many are now transitioning into retirement. We examined the histories of employment, retirement and disability status on depression among 1,497 Participants living with HIV from 1997 to 2015 in the Multicenter AIDS Cohort Study. Data were collected on depressive symptoms, employment, retirement, disability status as well as HIV-related and sociodemographic characteristics. Employment, retirement and disability status were lagged 2 years to assess whether the risk of depression at a given observation were temporally predicted by each respective status, adjusting for prior depressive symptoms and covariates. Being employed (aOR: 0.76; 95% CI: 0.71–0.82) had lower odds of depression risk two years later compared to those unemployed. There were higher odds of depression risk associated with disability (aOR: 1.43; 95% CI: 1.32–1.54) versus those not on disability. Retirement status was not associated with the risk of depressive symptoms. These findings could help inform policies and employment programs to facilitate the return to work for PLWH who are willing and able to work.

## Introduction

Prior to the advent of potent combination antiretroviral therapy (cART), persons living with HIV (PLWH) dealt with the consequences of an episodic illness that limit their ability to

of Confidentiality for investigator-specific concept sheet proposals. The data are available from the data coordinating center at JHU Bloomberg School of Public Health for researchers who meet the criteria for access to confidential data. The data underlying the results presented in the study are available from mwccs@jhu.edu.

**Funding:** The MWCCS is funded primarily by the National Heart, Lung, and Blood Institute (NHLBI), with additional co-funding from the Eunice Kennedy Shriver National Institute Of Child Health & Human Development (NICHD), National Institute On Aging (NIA), National Institute Of Dental & Craniofacial Research (NIDCR), National Institute Of Allergy And Infectious Diseases (NIAID), National Institute Of Neurological Disorders And Stroke (NINDS), National Institute Of Mental Health (NIMH), National Institute On Drug Abuse (NIDA), National Institute Of Nursing Research (NINR), National Cancer Institute (NCI), National Institute on Alcohol Abuse and Alcoholism (NIAAA), National Institute on Deafness and Other Communication Disorders (NIDCD), National Institute of Diabetes and Digestive and Kidney Diseases (NIDDK), National Institute on Minority Health and Health Disparities (NIMHD), and in coordination and alignment with the research priorities of the National Institutes of Health, Office of AIDS Research (OAR) [U01-HL146241, U01-HL146201, U01-HL146204, U01-HL146202, U01-HL146193, U01-HL146245, U01-HL146240, U01-HL146242, U01-HL146333, U01-HL146205, U01-HL146203, U01-HL146208, U01-HL146192, U01-HL146194, and U01-AI35042]. This study is also funded by the National Institute for Minority Health Disparities [R01 MD010680 Plankey & Friedman]. The funders had no role in study design, data collection and analysis, decision to publish, or preparation of the manuscript.

**Competing interests:** The authors have declared that no competing interests exist.

participate in the workforce [1–3]. Wunderlich et al reported an increase in disability benefit awards to individuals based to their HIV infection in the late 80s, supporting the notion that many PLWH were unable to work [4]. With advancements in the management of HIV, some PLWH were presented with the opportunity to return to the workforce and many now are beginning the transition into retirement [1, 3, 5].

Unemployment has been identified as an independent predictor of depression in general populations [6–8]. Conversely, a systematic review of longitudinal studies showed a beneficial effect of return to work on health in working-aged adults and in a variety of times and settings [9]. The health benefits associated with going back to work in general populations resulted from health improvements after reemployment or health declines associated with continued unemployment. Specifically, PLWH who are unable to work are at significantly increased risk for psychological distress, suicidal ideation, psychiatric symptoms, anxiety, and are more than twice as likely to be hospitalized or die [10–13]. Benefits of employment and working conditions among PLWH have also been documented. Participation in employment has been associated with better physical and mental health among PLWH [14]. The quality of employment —or the presence of adverse psychosocial work conditions, such as job insecurity, high psychological demands, and lack of decision authority—also matters to health and depressive symptoms, suggesting that "bad jobs" may not offer the same mental health advantages as "good jobs" [15, 16]. In a large-scale study of 2,864 PLWH, researchers found that those who were employed reported lower depression symptoms and increased social functioning than those who were not employed [17]. A similar study reported that employed PLWH better handled life difficulties, had lower psychological stress, and better managed their health than unemployed PLWH [18]. Despite the potential health advantages of employment, unemployment rates among PLWH remain higher than the general population [19–22].

Previous literature has been inconsistent in regards to the impact of retirement on mental health. Mandel and Roe (2008) reported improved mental health of older individuals who are retired [23]. Another study reported lower levels of depression during an individual's first year of retirement, and higher levels thereafter [24]. Retirement is correlated with decreased physical health, decreased mobility, and challenges with activities of daily living, which becoming increasing present with older age [25, 26]. Further, information regarding retirement and depression among PLHWA. Therefore, it is important to ascertain any potential impact of retirement on mental health among aging populations living with HIV.

While the relationship between employment, disability and retirement and depression has been established in the general population, information is limited among PLWH. Rueda [27] conducted the first US-based longitudinal study using data from the Multicenter AIDS Cohort Study (MACS) to investigate the relationship between employment and health-related quality of life among PLWH. While they found employment to be associated with better physical and mental quality of life over time, the study was not designed to examine a time-lagged temporal effect between employment status and mental health [27] nor the relationship between retirement and depression risk. We sought to investigate the impact of employment, retirement and disability history on future risk of depression among participants living with HIV in the Multicenter AIDS Cohort Study from 1997 to 2015. We decided to conduct time-lagged analyses to allow for the potential effects of employment, disability, and retirement on health outcomes to become salient during the observation period, and we also wanted to consider the potential adaptation to the experience of retirement or disability [28]. We hypothesize that employed and retired participants will have a lower depression risk while those on disability will have a higher depression risk, after controlling for clinical and sociodemographic covariates.

## Materials and methods

### Population

Since 1984, the Multicenter AIDS Cohort Study (MACS) has enrolled successive observational cohorts of men who have sex with men (MSM) with and without HIV in four US sites: Pittsburgh, Pennsylvania/Columbus, Ohio; Los Angeles, California; Chicago, Illinois; and Baltimore, Maryland /Washington, DC. Since its inception in 1984, a total of 7,352 participants have been enrolled in the study over 4 periods: 4,954 in 1984–1985; 668 in 1987–1991; 1350 in 2001–2003; 380 in 2011-current. MACS participants are seen at semiannual visits to undertake blood draws, physical exams, and behavioral and medical questionnaires. The MACS study design has been described elsewhere [29, 30]. Questionnaires are available at www. aidscohortstudy.org. Institutional review boards at John Hopkins University, Northwestern University, University of California Los Angeles and University of Pittsburgh approved the protocol, and written informed consent was obtained from all study participants. The current analyses center on employment, disability, and retirement status and depressive symptomatology collected from 1,497 MACS participants living with HIV over an 18-year period, between October 1997—September 2015 (study visits 28 and 63).

### Covariates

**Sociodemographics.** Each participant reported race/ethnicity and date of birth at their enrollment visit. Age was calculated at each visit and categorized as 18–29; 30–39; 40–49; 50–59; 60–69; and 70 years or older. Self-reported educational attainment was represented at a given observation as 8th grade or less; some high school but no high school graduation or GED; high school graduate/GED; some college but no college degree; college degree; some graduate work and post graduate degree.

**HIV status.** HIV status was assessed using enzyme-linked immunosorbent assay with confirmatory Western blot on all MACS participants at their initial visit and at every visit for those HIV-negative. Participants living with HIV included all men who were identified as such at baseline and those who seroconverted during study observation.

**Therapy.** Participants reported the type of antiretroviral therapy for their HIV infection used since last visit. The responses included no therapy, mono-therapy, combination therapy, and potent antiretroviral therapy (ART).

**Viral load.** Plasma HIV RNA levels (viral load, copies/ml) were collected. Viral load was dichotomized into detectable and undetectable, based on the lower detection level of the assay used at visit.

**Social support.** Participants were asked if they had someone to talk to or count on for social support. The responses were: 1) No, no one; 2) Yes, 1 person; 3) 2 or 3 people; 4) 4 or 5 people; and 5) 6 or more people. The variable was recoded into low (0–1 person), medium (2–3 people) and high (4 or more people) social support [31]. In this sample, higher social support was associated with lower levels of negative psychosocial health conditions, including depression [31]. Therefore, we adjusted the model in order to account for possible confounding effect of social support.

**Depression risk.** Depressive symptoms were assessed at each study visit using self-administered questions derived from symptom checklist developed by the Center for Epidemiologic Study of Depression. The response values ranged from 0 to 3. Values were summed at each observation for all participants who completed the 20-question measure. The score ranged from a minimum of 0 to a maximum of 60. We created a dichotomous variable assessing whether participants had a score ≥16, a cut-off point associated with the risk of depression [32].

**Employment, disability and retirement status.**   Employment ("Employed", "Not Employed") was derived from self-reported responses on full time, part time and self-employment status. A participant was considered "employed" if they reported any current full-time, part-time or self-employment. If the participant did not report any full-time, part-time or self-employment, they were considered to be "not employed". Missing values on full time, part time, and self-employment status were coded as missing. Disability and retirement status was self-reported by the participant. Employment, disability and retirement status were not mutually exclusive categories; therefore, each outcome was modeled separately. We explored potential overlap between participants reporting combinations of employment, disability and retirement status and found the prevalence of overlap across participant visits to be less than 6%, supporting our decision to model them separately.

**Missing values.**   Missing values were imputed by examining employment status in the prior visit and retaining the response from the prior observation. If employment status was not reported in the prior visit, we used the response from the following visit. If there were no values for employment status in the prior and following visit, then the employment status remained missing. Missing values on disability and retirement status were imputed in a similar manner.

## Statistical analysis

Descriptive statistics for the outcome variable and covariates were generated at the index (October 1997/Visit 28), last (September 2015/Visit 65), and across all visits using frequencies/percentages and medians/interquartile ranges (IQRs) where appropriate. To assess the temporal relationship between employment, retirement and disability status and depressive symptoms, we fit separate generalized linear mixed models with a repeated measures statement controlling for within-subject variance. We programmed a 2-year lag for employment, retirement and disability in order to assess if depression risk at a given observation was predicted by these predictor variables at earlier study visits because the potential effects of labor market participation on health takes time to develop [28]. Additionally, we lagged the risk depression 2 years to adjust for prior risk of depression while predicting the future risk of depression. Analyses were conducted using SAS PROC GLIMMIX (SAS version 9.3, SAS Institute Inc., Cary, NC, USA). We assessed separately whether the risk of depression (CES-D score $\geq$16) at a given observation were predicted respectively by employment, disability, and retirement, adjusting for time (study visit), age, race/ethnicity, education, social support, risk of depression 2 years prior, antiretroviral therapy and viral load detectability. We reported adjusted odds ratios (aOR) and 95% confidence intervals (CI).

## Results

### Population characteristics

There were 1,497 participants living with HIV in the analysis. Characteristics of the population was reported at the index visit (28), last visit (63) and across all visits (28–63) (Table 1). The majority of participants identified as White, non-Hispanic (53.2%), was between 30 and 50 years old (69.3%) and had at least a high school diploma (92.0%) at their first visit. Across all visits, most reported using potent ART to treat their HIV infections (75.0%), maintained an undetectable viral load (73.2%), and had a medium (40.0%) or high (34.5%) level of social support. There were 3,026 participant-visits with missing employment status. After imputation procedures, there were 157 participant-visits with missing employment status. Risk of depression was seen in 27.5% of the participants. Participants reported employment, retirement and disability at 64.7%, 9.4% and 23.8% across all visits, respectively.

**Table 1. Characteristics of study participants.**

| | Index Visit (28) | Last Visit (63) | All Visits (28–63) |
|---|---|---|---|
| | n = 1497 | n = 1117 | N = 31489 |
| **Race/Ethnicity** | | | |
| White, non-Hispanic | 797 (53.2%) | 604 (54.1%) | 19129 (60.8%) |
| Hispanic | 233 (15.6%) | 176 (15.8%) | 3901 (12.4%) |
| Black, non-Hispanic | 439 (29.3%) | 318 (28.5%) | 8094 (25.7%) |
| Other | 28 (1.9%) | 19 (1.7%) | 364 (1.2%) |
| **Age** | | | |
| 17–29 years old | 159 (10.6%) | 42 (3.8%) | 722 (2.3%) |
| 30–39 years old | 481 (32.1%) | 104 (9.3%) | 3353 (10.7%) |
| 40–49 years old | 557 (37.2%) | 171 (15.3%) | 10042 (31.9%) |
| 50–59 years old | 141 (9.4%) | 387 (34.6%) | 10267 (32.6%) |
| 60–69 years old | 22 (1.5%) | 279 (25.0%) | 3474 (11.0%) |
| 70 + years old | 137 (9.2%) | 134 (12.0%) | 3631 (11.5%) |
| **Education** | | | |
| 8th grade or less | 30 (2.0%) | 18 (1.6%) | 530 (1.7%) |
| 9, 10, 11th grade | 89 (6.0%) | 62 (5.6%) | 1440 (4.6%) |
| 12th grade | 253 (16.9%) | 184 (16.5%) | 4673 (14.8%) |
| At least one year of college, but no degree | 477 (31.9%) | 344 (30.8%) | 9829 (31.2%) |
| Four years of college/Obtained Degree | 318 (21.2%) | 253 (22.7%) | 7001 (22.2%) |
| Some graduate work | 119 (8.0%) | 85 (7.6%) | 2941 (9.3%) |
| Post-graduate degree | 211 (14.1%) | 171 (15.3%) | 5074 (16.1%) |
| **Therapy** | | | |
| No Therapy | 585 (40.1%) | 50 (4.5%) | 4909 (15.8%) |
| Mono Therapy | 82 (5.6%) | 7 (0.6%) | 145 (0.5%) |
| Combination Therapy | 229 (15.7%) | 27 (2.5%) | 2701 (8.7%) |
| Potent ART | 562 (38.6%) | 1017 (92.4%) | 23237 (75.0%) |
| Missing | 39 | 16 | 497 |
| **Social Support** | | | |
| Low | 369 (26.5%) | 284 (27.7%) | 7257 (25.5%) |
| Medium | 552 (39.7%) | 418 (40.8%) | 11372 (40.0%) |
| High | 471 (33.8%) | 323 (31.5%) | 9790 (34.5%) |
| Missing | 105 | 92 | 3070 |
| **Viral Load** | | | |
| Undetectable | 530 (40.5%) | 877 (90.1%) | 20000 (73.2%) |
| Detectable | 780 (59.5%) | 96 (9.9%) | 7336 (26.8%) |
| Missing | 187 | 144 | 4153 |
| **Disability** | | | |
| Yes | 251 (18.0%) | 238 (22.7%) | 6799 (23.8%) |
| No | 1140 (82.0%) | 809 (77.3%) | 21824 (76.3%) |
| Missing | 106 | 70 | 2866 |
| **Retired** | | | |
| Yes | 41 (3.0%) | 161 (15.4%) | 2699 (9.4%) |
| No | 1350 (97.1%) | 886 (84.6%) | 25924 (90.6%) |
| Missing | 106 | 70 | 2866 |
| **Employed** | | | |
| Yes | 1033 (69.5%) | 666 (59.9%) | 20263 (64.7%) |
| No | 454 (30.5%) | 446 (40.1%) | 11069 (35.3%) |

(*Continued*)

**Table 1.** (Continued)

|  | Index Visit (28) | Last Visit (63) | All Visits (28–63) |
|---|---|---|---|
|  | n = 1497 | n = 1117 | N = 31489 |
| Missing | 10 | 5 | 157 |
| **Depression** |  |  |  |
| Yes | 440 (32.3%) | 279 (27.3%) | 7748 (27.5%) |
| No | 922 (67.7%) | 743 (72.7%) | 20477 (72.6%) |
| Missing | 135 | 95 | 3264 |

## Employment, retirement and disability status on the risk of depression

After adjusting for race/ethnicity, age, education, prior risk of depression, social support, and viral load, being employed was associated with a 24% decrease in odds of depression risk compared to being unemployed (aOR: 0.73; 95% CI: 0.68–0.78) (Table 2). Participants reporting disability had increased odds of depression risk (aOR: 1.43; 95% CI: 1.32–1.54) (Table 3). Retirement was not statistically associated with depression risk (Table 4).

## Other factors associated with depression risk

Compared to MSM 70 years and older, younger participants had increased depression risk: 17–29 years (aOR: 2.18; 95% CI: 1.41–3.37); 30–39 years (aOR: 2.27; 95% CI: 1.54–3.35); 40–49 years (aOR: 2.38; 95% CI: 1.62–3.47); 50–59 years (aOR: 2.13; 95% CI: 1.46–3.11); 60–69 years (aOR: 0.81; 95% CI: 1.09–2.35);. Lower educational attainment was associated with higher odds of depression risk compared to having a post-graduate degree: 8th grade or less (aOR: 1.52; 95% CI: 1.18–1.96); 9-11th grade (aOR: 1.84; 95% CI: 1.55–2.19); 12th Grade (HS diploma) (aOR: 1.39; 95% CI 1.22–1.58); some college (aOR: 1.45; 95% CI: 1.30–1.62); and college degree (aOR: 1.24; 95% CI: 1.10–1.40). Compared to having a medium level of social support, having high social support had lower odds of depression risk (aOR: 0.62; 95% 0.57–0.68), while low social support was associated with higher odds of depression risk (aOR: 1.98; 95% CI: 1.83–2.15). An undetectable viral load was protective against depression risk (aOR: 0.69; 95% CI: 0.63–0.76) (Table 2). Covariates in the disability and retirement models were similar in magnitude and statistical significance and are detailed in Tables 3 and 4, respectively.

## Discussion

This paper aims to improve our understanding of the complex link between work and health in men living with HIV by conducting time-lagged longitudinal analyses to account for the delayed effects of upstream determinants of health on health outcomes and the potential adaptations to the experiences of employment, retirement, and disability. Following a social determinants of health approach, these findings contribute to the study of the causation hypothesis —employment leads to better health among people with HIV. The selection hypothesis— health is a necessary condition for employment—was outside the scope of this paper but cannot be discounted (although previous studies have suggested that the magnitude of the causation effect may be larger than the selection effect) [14, 33, 34]. Contemporary thinking on these competing hypotheses, however, has been moving away from such binary distinctions to examining how both processes work in tandem to shape health and illness trajectories over time.

This longitudinal study found that employed men living with HIV had a lower likelihood of depression risk over time, after adjusting for key clinical and sociodemographic variables. This

**Table 2. Longitudinal associations between employment and risk of depression (participant N = 1482; total observation N = 29970).**

|  | aOR | 95% Confidence Interval | p-value |
|---|---|---|---|
| **2 year Lag Employment** |  |  |  |
| Employed | 0.73 | 0.68–0.78 | <0.0001 |
| Not Employed (Referent) | - | - | - |
| **Prior Risk of Depression (2-year Lag)** |  |  |  |
| Risk of depression | 6.20 | 5.79–6.63 | <0.0001 |
| No risk of depression (Referent) | - | - | - |
| **Race/Ethnicity** |  |  |  |
| Black | 0.99 | 0.91–1.08 | 0.6815 |
| Hispanic | 0.92 | 0.83–1.03 | 0.6214 |
| Other | 1.08 | 0.81–1.44 | 0.4419 |
| White (Referent) | - | - | - |
| **Age** |  |  |  |
| 17–29 years old | 2.18 | 1.41–3.37 | 0.0003 |
| 30–39 years old | 2.27 | 1.54–3.35 | <0.0001 |
| 40–49 years old | 2.38 | 1.62–3.47 | <0.0001 |
| 50–59 years old | 2.13 | 1.46–3.11 | <0.0001 |
| 60–69 years old | 1.60 | 1.09–2.35 | 0.0097 |
| 70 + years old (Referent) | - | - | - |
| **Education** |  |  |  |
| ≤8th grade or less | 1.52 | 1.18–1.96 | 0.0014 |
| 9, 10, 11th grade | 1.84 | 1.55–2.19 | <0.0001 |
| 12th grade | 1.39 | 1.22–1.58 | <0.0001 |
| At least one year of college, but no degree | 1.45 | 1.30–1.62 | <0.0001 |
| Four years of college/Obtained degree | 1.24 | 1.10–1.40 | 0.0004 |
| Some graduate work | 1.14 | 0.98–1.33 | 0.0927 |
| Post-graduate degree (Referent) | - | - | - |
| **Social Support** |  |  |  |
| High | 0.62 | 0.57–0.68 | <0.0001 |
| Low | 1.98 | 1.83–2.15 | <0.0001 |
| Medium (Referent) | - | - | - |
| **Therapy** |  |  |  |
| None | 0.81 | 0.72–0.91 | 0.0006 |
| Mono | 1.02 | 0.64–1.63 | 0.9373 |
| Combo | 1.16 | 1.02–1.32 | 0.0208 |
| Potent (Referent) | - | - | - |
| **Viral Load** |  |  |  |
| Undetectable | 0.69 | 0.63–0.76 | <0.0001 |
| Detectable (Referent) | - | - | - |

finding is in agreement with previous studies that have examined the cross-sectional and longitudinal associations between employment and mental health in people with HIV, including depressive symptoms [14, 15, 35]. It also extends previous analyses conducted with MACS data by introducing a two-year lag between the measurement of employment status and the assessment of the mental health outcome. In this previous MACS paper we examined the impact of employment on physical and mental health quality of life while the present paper examines the effects of employment on depression risk, a more direct measure of mental

**Table 3. Longitudinal associations between disability and risk of depression (participant N = 1418; total observation N = 27367).**

| | aOR* | 95% Confidence Interval | p-value |
|---|---|---|---|
| **2 year Lag Disability** | | | |
| Disabled | 1.43 | 1.32–1.54 | <0.0001 |
| Not Disabled (Referent) | - | - | - |
| **Prior Risk of Depression (2 year Lag)** | | | |
| Risk of depression | 6.20 | 5.80–6.63 | <0.0001 |
| No risk of depression (Referent) | - | - | - |
| **Race/Ethnicity** | | | |
| Black | 0.99 | 0.91–1.08 | 0.8163 |
| Hispanic | 0.94 | 0.84–1.05 | 0.2383 |
| Other | 1.13 | 0.85–1.51 | 0.4073 |
| White (Referent) | - | - | - |
| **Age** | | | |
| 17–29 years old | 1.86 | 1.21–2.88 | 0.0051 |
| 30–39 years old | 1.91 | 1.30–2.81 | 0.0011 |
| 40–49 years old | 1.97 | 1.35–2.87 | 0.0005 |
| 50–59 years old | 1.75 | 1.20–2.55 | 0.0035 |
| 60–69 years old | 1.37 | 0.94–2.02 | 0.1053 |
| 70 + years old (Referent) | - | - | - |
| **Education** | | | |
| ≤8th grade or less | 1.44 | 1.12–1.86 | 0.0049 |
| 9, 10, 11th grade | 1.76 | 1.48–2.09 | <0.0001 |
| 12th grade | 1.29 | 1.14–1.47 | <0.0001 |
| At least one year of college, but no degree | 1.36 | 1.22–1.52 | <0.0001 |
| Four years of college/Obtained degree | 1.20 | 1.07–1.35 | 0.0015 |
| Some graduate work | 1.08 | 0.93–1.25 | 0.313 |
| Post-graduate degree (Referent) | - | - | - |
| **Social Support** | | | |
| High | 3.25 | 2.98–3.55 | <0.0001 |
| Low | 1.62 | 1.49–1.76 | <0.0001 |
| Medium (Referent) | - | - | - |
| **Therapy** | | | |
| None | 0.87 | 0.78–0.97 | 0.0155 |
| Mono | 0.97 | 0.65–1.47 | 0.9 |
| Combo | 1.04 | 0.93–1.17 | 0.475 |
| Potent (Referent) | - | - | - |
| **Viral Load** | | | |
| Undetectable | 0.72 | 0.66–0.78 | <0.0001 |
| Detectable (Referent) | - | - | - |

health. The present time-lagged finding further supports the temporal sequence between change in employment status and the mental health outcome because the results of the previous longitudinal analysis incorporated both the between-subject effects (i.e., the difference between employed and unemployed individuals) and the within-subject effects (i.e., the within individual change in mental health due to change in employment status) [27]. Those regression coefficients estimated the difference in mental health scores for the overall population of employed people compared to the overall population of unemployed people. Such "pooling" of

**Table 4. Longitudinal associations between retirement and risk of depression (participant N = 1389; total observation N = 24129).**

| | aOR[*] | 95% Confidence Interval | p-value |
|---|---|---|---|
| **2 year Lag Retirement** | | | |
| Retired | 1.10 | 0.96–1.26 | 0.1869 |
| Not Retired (Referent) | - | - | - |
| **Prior Risk of Depression (2 year Lag)** | | | |
| Risk of depression | 6.49 | 6.07–6.94 | <0.0001 |
| No risk of depression (Referent) | - | - | - |
| **Race/Ethnicity** | | | |
| Black | 1.04 | 0.96–1.13 | 0.3339 |
| Hispanic | 0.93 | 0.83–1.03 | 0.1703 |
| Other | 1.07 | 0.80–1.43 | 0.6558 |
| White (Referent) | - | - | - |
| **Age** | | | |
| 17–29 years old | 1.98 | 1.27–3.10 | 0.0028 |
| 30–39 years old | 2.11 | 1.42–3.16 | 0.0003 |
| 40–49 years old | 2.21 | 1.49–3.27 | <0.0001 |
| 50–59 years old | 1.98 | 1.35–2.93 | 0.0006 |
| 60–69 years old | 1.54 | 1.04–2.27 | 0.0313 |
| 70 + years old (Referent) | - | - | - |
| **Education** | | | |
| ≤8th grade or less | 1.61 | 1.25–2.07 | 0.0002 |
| 9, 10, 11th grade | 1.91 | 1.61–2.27 | <0.0001 |
| 12th grade | 1.39 | 1.22–1.58 | <0.0001 |
| At least one year of college, but no degree | 1.42 | 1.27–1.58 | <0.0001 |
| Four years of college/Obtained degree | 1.23 | 1.10–1.38 | 0.0005 |
| Some graduate work | 1.09 | 0.94–1.27 | 0.2344 |
| Post-graduate degree (Referent) | - | - | - |
| **Social Support** | | | |
| High | 0.63 | 0.58–0.68 | <0.0001 |
| Low | 2.02 | 1.87–2.19 | <0.0001 |
| Medium (Referent) | - | - | - |
| **Therapy** | | | |
| None | 0.85 | 0.76–0.95 | 0.0046 |
| Mono | 0.98 | 0.65–1.48 | 0.9142 |
| Combo | 1.06 | 0.94–1.19 | 0.3639 |
| Potent (Referent) | - | - | - |
| **Viral Load** | | | |
| Undetectable | 0.71 | 0.65–0.77 | <0.0001 |
| Detectable (Referent) | - | - | - |

cross-sectional and longitudinal relationships made it impossible to isolate the relative contribution of each. However, the present analyses define a temporal sequence by introducing a time-lag between the measure of employment and the measure of depression risk, which offers some reassurances that the change in mental health is better characterized as a result of the employment transition. The importance of conducting time-lagged analyses is supported in part by set-point theories, which propose that people's experiences to stressful events show a decline in well-being only to return to baseline over time. This observation that people may

experience adaptations to stressful experiences has been documented in some studies but not consistently [36–38].

Our study also found out that participants on disability had an increased odds of depression risk after controlling for prior depression and other covariates. In the United States, persons with depression may be eligible for social security disability income [39]. Therefore, the relationship between depression and disability status can be bidirectional: 1) depression as a criterion for disability; and 2) disability as the cause of depression. By including pre-existing depression, we were able to tease out a causal relationship of disability on depression risk.

Previous studies regarding the association between retirement and depressive symptoms in general populations have been equivocal and the study of the health effects of retirement on people with HIV has so far been neglected in the literature. Greenfield et al found that, during retirement, increased social activities correlated with decreased depressive symptoms [40]. Reitzes (1996) noted less depression in the first year of retirement, but increasing depression in subsequent years [24]. In a 10-year cross-aged analysis, Segel-Karpas examined the reciprocal effects of retirement and depression. They found that retirement increased the likelihood of depressive symptoms and depression symptoms increased the inclination to retire [28]. In our analysis, the relationship between retirement and depression risk was nonsignificant after adjusting for prior existence of depressive symptoms.

The strength of this study includes the longitudinal investigation of the depression risk associated with employment and disability—and to our knowledge for the first time extends this work to the experience of retirement—in a well characterized and large cohort of men who have sex with men and living with HIV. We also followed an analytical approach that included a time-lag to establish a temporal sequence between our predictors (employment, disability, and retirement) and depression risk, with a comprehensive adjustment for a range of potential confounders. This allowed us to take into account the potential delayed effects of upstream determinants of health and the experience of adaptation to different stressors such as unemployment, disability, and retirement. An important limitation of this study relates to residual confounding because we did not have data on important variables, especially those related to income, reasons for retirement and disability, length of time on disability or retirement, and other systemic barriers to employment, retirement and disability programs such as information on disability benefits, unemployment insurance, or work history. Our analyses however controlled for a number of important covariates related to the experience of living with HIV and labor force participation, including race/ethnicity, age, education, social support, antiretroviral therapy, and viral load.

In addition, our conceptual framework for the selection of the predictors and outcomes focused on the determinants of health and not on selection effects. It is entirely possible that experiencing higher depressive symptoms may interfere with participants' ability to remain employed, increase the risk of going into disability, or precipitate early retirement. Future research should examine the relative contribution of causation versus selection effects and how they interact and reinforce each other as both have different clinical and policy implications. Due to limited sample size, we were unable to explore differences between participants who retired due to disability and those who retired without disability as this would have allowed us to examine important distinctions in the experience of retirement and its effects on subsequent health outcomes. Lastly, this study was restricted to MSM with HIV in four major metropolitan areas; therefore, results may not be generalizable to the general population of people living with HIV.

This research provided further evidence that employment leads to mental health benefits, and for the first time also suggests that retirement is also associated with better mental health outcomes, which suggests that policies that focus on improving employment opportunities

and better retirement conditions may have a significant impact on the health of men living with HIV. It is well known that better disability policies may reduce some of the negative financial conditions people with HIV experience and the stress associated with maintaining employment, remaining on disability benefits, and a successful transition to retirement. However, this work in progress needs to be strengthened by further research to develop and support sensible clinical and policy recommendations to support people living with HIV. Policy research is needed to identify and address structural issues in the benefits and drug coverage programs that may be implicated in creating barriers for people with HIV to return to work or remained employed, provide additional supports while on disability (such as mental health treatment), and facilitate a successful transition to retirement when needed.

## Acknowledgments

The authors are indebted to the participants of the Multicenter AIDS Cohort Study [MACS] Healthy Aging Study. The authors thank the staff at the four sites for implementation support John Welty, Montserrat Tarrago, and Katherine McGowan for data support of this study.

## Author Contributions

**Conceptualization:** Michael Plankey.

**Formal analysis:** Deanna Ware, M. Reuel Friedman.

**Methodology:** Deanna Ware.

**Supervision:** Michael Plankey.

**Writing – original draft:** Deanna Ware, Sergio Rueda, Michael Plankey.

**Writing – review & editing:** Sergio Rueda, Michael Plankey, Pamela Surkan, Chukwuemeka N. Okafor, Linda Teplin, M. Reuel Friedman.

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
