## [Decision Letter · Decision Letter 0]

6 Aug 2020

PONE-D-20-14823

The longitudinal impact of employment, retirement and disability status on depressive symptoms in HIV-positive men in the Multicenter AIDS Cohort Study

PLOS ONE

Dear Dr. Ware,

Thank you for submitting your manuscript to PLOS ONE. After careful consideration, we feel that it has merit but does not fully meet PLOS ONE’s publication criteria as it currently stands. Therefore, we invite you to submit a revised version of the manuscript that addresses the points raised during the review process.

From my own reading of the manuscript, I agree with the reviewers comments (below). Please carefully consider each of the comments and incorporate as appropriate. I look forward to receiving your revision soon.

We look forward to receiving your revised manuscript.

Kind regards,

Ethan Morgan

Academic Editor

PLOS ONE

Journal Requirements:

"The MACS is primarily funded by the National 342 Institute of Allergy and Infectious

Diseases, with additional co-funding from the National Cancer Institute, the National Institute on

Drug Abuse, and the National Institute of Mental Health. Targeted supplemental funding for

specific projects was also provided by the National Heart, Lung, and Blood Institute and the

National Institute on Deafness and Communication Disorders. MACS data collection is also

supported by grant UL1-TR000424 (Johns Hopkins University Institute for Clinical and

Translational Research) from the National Center for Advancing Translational Sciences, a

component of the National Institutes of Health (NIH), and NIH Roadmap for Medical Research.

This research was supported by the NIH via interagency agreement with the National Institute of

Allergy and Infectious Diseases, the Eunice Kennedy Shriver National Institute of Child Health

and Human Development, and other NIH Cooperative Agreements (U01-HD-32632):

Disclaimer: The contents of this publication are solely the responsibility of the authors and do

not represent the official views of the Johns Hopkins Institute for Clinical and Translational

Research, National Center for Advancing Translational Sciences, NIH, the Department of Health

and Human Services, or the US government."

"The authors received no specific funding for this work."

Reviewers' comments:

Reviewer's Responses to Questions

**Comments to the Author**

1. Is the manuscript technically sound, and do the data support the conclusions?

Reviewer #1: Yes

Reviewer #2: Partly

2. Has the statistical analysis been performed appropriately and rigorously? 

Reviewer #1: Yes

Reviewer #2: Yes

3. Have the authors made all data underlying the findings in their manuscript fully available?

Reviewer #1: Yes

Reviewer #2: Yes

4. Is the manuscript presented in an intelligible fashion and written in standard English?

Reviewer #1: Yes

Reviewer #2: Yes

5. Review Comments to the Author

Reviewer #1: Overall, this study utilizes a well-known and conducted research cohort, yet still manages to assess a novel question that will contribute to current literature. This is a very good paper with well-written and thought out analyses and results. The discussion is well developed and appropriately follows the findings. I only had a few suggestions below for the authors to consider.

1. I would suggest removing retirement findings from the results as they are non-significant, or at least moving it to the end of the paragraph after the significant findings.

2. Any hypothesis as to why those 60-69 had reduced odds of depression risk?

3. I would suggest including N’s for the models in each of table 2, 3, and 4

Reviewer #2: I commend the authors for seeking to look at healthy aging and employment among men living with HIV. This is a robust data set and provides ample opportunity to examine changes in labor engagement and mental health across the lifespan. I believe this manuscript has potential to make an important contribution and I have outlined suggested revisions and concerns below.

Comments:

1. The paragraph in the introduction on benefits/risks of retirement feels a bit disjointed. Revisiting this paragraph to clarify the primary point with better flow between supporting (or conflicting) studies would make it clearer why studying retirement specifically is important.

2. In line 75, participants are referred to as HIV-positive men. Elsewhere the preferred person-first language “People living with HIV” is used. Here, I suggest changing the language to “men living with HIV” in this sentence. People first, non-stigmatizing language should be used throughout. HIV-positive status can be changed to HIV status. HIV positive participants can be changed to participants living with HIV.

3. Social support is identified as a covariate but it isn’t explained why this construct would be important to include in the methods or introduction but it is mentioned briefly in the discussion. I suggest making it more clear why this is included and potentially discussing it’s link as an implied mechanism for linking mental health and labor engagement.

4. More detail about the depression measure would be helpful. How could a participant achieve a score of 16 to exceed the clinical threshold? For a reader to know this, the scoring on individual items should be reported. For example, on the CESD items are typically rated by frequency ranging from 0 to 3.

5. How many participants required imputation for missing employment status? Were any sensitivity checks completed to see if the imputation approach to missingness changed results? Similarly, why was this approach used for employment but not disability and retirement?

6. How was the index visit chosen and what does this mean?

7. The authors make the point (without citation) that the link between labor related determinants and depression is expected to be lagged, however, it is unclear why a 2 year lag was chosen. At the aggregate level it appears justified that average sample levels of depression may lag behind major labor market shifts quite slowly but it’s less clear why unemployment would take two years to impact mental health, particularly if in that two years employment status changes again. Because conclusions are drawn about implied causality, it is important to justify this time lag. There may be a number of confounding events or factors in that two year window that could also contribute to mental health.

8. The authors refer to both within and between subjects differences in the methods, however, no within subjects analysis was conducted. The repeated measures statement in SAS typically uses subject ID to indicate within the model that some data points belong to the same participant and are thus non-independent. Repeated measures from multiple subjects do provide more robust data however it appears only between subjects results are reported in this study. Some clarification would help to make it clear that the results are between-subjects only.

9. It should be stated in text that retirement was not associated with odds of depression risk. The confidence band include 1.0 and the p-value is far from the alpha = .05 significance or even p = .10 marginal significance. Likewise in the discussion it is inappropriate to draw conclusions that retirement is associated with better mental health. This is not supported by the study results.

10. How might retirement status be confounded with age? Perhaps a different study design would be more appropriate for examining the impact of retirement. That is, younger individuals are highly unlikely to retire thus providing little variance in retirement data earlier in life. A within-person analysis would better answer the question whether an individual’s mental health is different when they retire. The introduction appeared to frame the paper, in part, around aging and retirement but the methods didn’t really focus on this age group or life transition. I do agree with authors that understanding how retirement is related to health is important in the aging cohort of men living with HIV, but don’t feel this paper really provides much insight into that process.

Editorial notes:

There are some minor editorial changes needed to ensure the same and correct tense is used across sections (for example, hypotheses should be in past tense).

6. PLOS authors have the option to publish the peer review history of their article (what does this mean?). If published, this will include your full peer review and any attached files.

Reviewer #1: No

Reviewer #2: No

---

## [Author Response · Author response to Decision Letter 0]

24 Aug 2020

Response to Reviewer #1:

Thank you for your review of our paper. We have answered each of your points below. Additionally, we performed missing value imputation for disability and retirement status. All models (employment, disability, and retirement) has been re-analyzed. Therefore, results vary slightly from the initial manuscript.

Reviewer # 1 Comment 

 I would suggest removing retirement findings from the results as they are non-significant, or at least moving it to the end of the paragraph after the significant findings. 

Author Response

Thank you for your comment. I moved the non-significant result of retirement to the end of the other significant findings as suggested.

Reviewer # 1 Comment 

Any hypothesis as to why those 60-69 had reduced odds of depression risk?

Author Response

I performed missing value imputation for retirement and disability variables. The models for employment, disability and retirement were then re-ran. Those 60-69 no longer have a statistically significant decreased odds of depression risk in all three models. 

Reviewer # 1 Comment 

 I would suggest including N’s for the models in each of table 2, 3, and 4 

Author Response

I added the participant N and visit level N into Tables 2, 3 and 4 as suggested.

Response to Reviewer #2:

Thank you for your review of our paper. We have answered each of your points below. Additionally, we performed missing value imputation for disability and retirement status. All models (employment, disability, and retirement) has been re-analyzed. Therefore, results vary slightly from the initial manuscript.

Reviewer # 2 Comment 

The paragraph in the introduction on benefits/risks of retirement feels a bit disjointed. Revisiting this paragraph to clarify the primary point with better flow between supporting (or conflicting) studies would make it clearer why studying retirement specifically is important. 

Author Response 

Thank you for your comment. I made changes to the paragraph introducing retirement for clarity (see lines 59-67) as suggested.

Reviewer # 2 Comment 

In line 75, participants are referred to as HIV-positive men. Elsewhere the preferred person-first language “People living with HIV” is used. Here, I suggest changing the language to “men living with HIV” in this sentence. People first, non-stigmatizing language should be used throughout. HIV-positive status can be changed to HIV status. HIV positive participants can be changed to participants living with HIV.

Author Response 

As suggested, we used person first language throughout the manuscript.

Reviewer # 2 Comment 

Social support is identified as a covariate but it isn’t explained why this construct would be important to include in the methods or introduction but it is mentioned briefly in the discussion. I suggest making it more clear why this is included and potentially discussing it’s link as an implied mechanism for linking mental health and labor engagement. 

Author Response 

Friedman et al assessed social support in this sample of men. They found that higher social support was associated with lower levels of negative psychosocial health outcomes (which includes depression). This has been added to the manuscript to make it clear on why we chose to include it as a covariate in the models (see lines 118 to 121).

Reviewer # 2 Comment 

More detail about the depression measure would be helpful. How could a participant achieve a score of 16 to exceed the clinical threshold? For a reader to know this, the scoring on individual items should be reported. For example, on the CESD items are typically rated by frequency ranging from 0 to 3 

Author Response 

As suggested, I added the value range for the responses and the minimum and maximum scores (See lines 124-127).

Reviewer # 2 Comment 

How many participants required imputation for missing employment status? Were any sensitivity checks completed to see if the imputation approach to missingness changed results? Similarly, why was this approach used for employment but not disability and retirement? 

Author Response 

There are 4,417 participant-visits with missing employment, retirement and disability.

Not imputing retirement and disability was an oversight as we were initially focused on employment status. Therefore, we have imputed the missing retirement and disability values. To ensure results did not change due to imputation, we compared the model results from imputed vs non-imputed data for each of the primary outcomes: employment, retirement and disability. The estimates for the primary predictors was less than 0.001. Therefore, we can confidently state that imputing missing values did not change our results.

Reviewer # 2 Comment 

How was the index visit chosen and what does this mean? 

Author Response 

Index visit is the first visit in our retroactive observation of employment, disability, retirement status and depressive symptoms.

We chose the index visit of October 1997 because it was the peak of the HAART initiation within the MACS. 

Many persons living with HIV were unable to work due to disease progression. By looking at the time periods in which potent HAART was introduced, we can assess how employment trends as PLWH become healthier and possibly re-enter the workforce.

Reviewer # 2 Comment 

The authors make the point (without citation) that the link between labor related determinants and depression is expected to be lagged, however, it is unclear why a 2 year lag was chosen. At the aggregate level it appears justified that average sample levels of depression may lag behind major labor market shifts quite slowly but it’s less clear why unemployment would take two years to impact mental health, particularly if in that two years employment status changes again. Because conclusions are drawn about implied causality, it is important to justify this time lag. There may be a number of confounding events or factors in that two year window that could also contribute to mental health. 

Author Response 

Segel-Karpas et al used a 2-year lag on employment status and depressive symptom and decided to use a similar approach. Lines 85 to 90 explains our reasoning for the 2-year lag.

“We decided to conduct time-lagged analyses to allow for the potential effects of employment, disability, and retirement on health outcomes to become salient during the observation period, and we also wanted to consider the potential adaptation to the experience of retirement or disability”.

Reviewer # 2 Comment 

The authors refer to both within and between subjects differences in the methods, however, no within subjects analysis was conducted. The repeated measures statement in SAS typically uses subject ID to indicate within the model that some data points belong to the same participant and are thus non-independent. Repeated measures from multiple subjects do provide more robust data however it appears only between subjects results are reported in this study. Some clarification would help to make it clear that the results are between-subjects only. 

Author Response 

We adjusted for between and within subject variances in our models. We did not report within subject differences. We will make it clear in the methods that only between subject results are reported (see line 160).

Reviewer # 2 Comment 

It should be stated in text that retirement was not associated with odds of depression risk. The confidence band include 1.0 and the p-value is far from the alpha = .05 significance or even p = .10 marginal significance. Likewise in the discussion it is inappropriate to draw conclusions that retirement is associated with better mental health. This is not supported by the study results. 

Author Response 

We have made the change to reflect that retirement was not associated with the risk of depressive symptoms.

Reviewer # 2 Comment 

 How might retirement status be confounded with age? Perhaps a different study design would be more appropriate for examining the impact of retirement. That is, younger individuals are highly unlikely to retire thus providing little variance in retirement data earlier in life. A within-person analysis would better answer the question whether an individual’s mental health is different when they retire. The introduction appeared to frame the paper, in part, around aging and retirement but the methods didn’t really focus on this age group or life transition. I do agree with authors that understanding how retirement is related to health is important in the aging cohort of men living with HIV, but don’t feel this paper really provides much insight into that process. 

Author Response 

This is a repeated measures analysis so we adjust for with-in person variance of the predictor variables over time. That said, not all men started at the same age—so we adjusted for age for retirement (for example, to account for early vs. late retirement).

---

## [Editor Report · Decision Letter 1]

3 Sep 2020

The longitudinal impact of employment, retirement and disability status on depressive symptoms among men living with HIV in the Multicenter AIDS Cohort Study

PONE-D-20-14823R1

Dear Dr. Ware,

We’re pleased to inform you that your manuscript has been judged scientifically suitable for publication and will be formally accepted for publication once it meets all outstanding technical requirements.

Kind regards,

Ethan Morgan

Academic Editor

PLOS ONE
---

## [Editor Report · Acceptance letter]

9 Sep 2020

PONE-D-20-14823R1 

The longitudinal impact of employment, retirement and disability status on depressive symptoms among men living with HIV in the Multicenter AIDS Cohort Study 

Dear Dr. Ware:

I'm pleased to inform you that your manuscript has been deemed suitable for publication in PLOS ONE. Congratulations! Your manuscript is now with our production department. 

Kind regards, 

on behalf of

Dr. Ethan Morgan 

Academic Editor

PLOS ONE